# The Growing Significance of microRNAs in Osteoporosis

**DOI:** 10.3390/cells14231905

**Published:** 2025-12-01

**Authors:** Alika Sarkar, Sana Sarkar, Afreen Anwar, Ji Woong Kim, Jae-Hyuck Shim, Aijaz Ahmad John

**Affiliations:** 1Gene Regulation Section, Laboratory of Molecular Biology and Immunology, National Institute of Aging, Baltimore, MD 21224, USA; 2Department of Neurology, The Johns Hopkins University, Baltimore, MD 21205, USA; 3Department of Biology and Biotechnology, Worcester Polytechnic Institute, 100 Institute Road, Worcester, MA 01609, USA; afreenanwar2712@gmail.com; 4Department of Genetic and Cellular Medicine, University of Massachusetts Chan Medical School, Worcester, MA 01655, USA; 5Horae Gene Therapy Center, University of Massachusetts Chan Medical School, Worcester, MA 01655, USA; 6Li Weibo Institute for Rare Diseases Research, University of Massachusetts Chan Medical School, Worcester, MA 01655, USA

**Keywords:** osteoporosis, miRNA, microRNA, bone remodeling, therapeutics

## Abstract

Osteoporosis is an aging-related disease characterized by low bone mineral density and deteriorated bone structure, resulting in an increased risk of fractures. Currently, most osteoporosis therapies target osteoclasts to inhibit bone resorption, while the three FDA-approved anabolic agents include parathyroid hormone, parathyroid hormone-related protein, and anti-sclerostin antibody that promote osteoblast function. However, long-term treatment with these agents is associated with potential adverse effects and decreased therapeutic efficacy. This has prompted exploration of novel therapeutic strategies, including microRNAs (miRNAs), which are emerging as promising candidates. miRNAs have been reported to play important roles in regulating pathways involved in bone formation and resorption. In addition to their direct roles in osteoblasts and osteoclasts, miRNAs also serve as key mediators of communication between these cells, which is essential for maintaining bone homeostasis. The complexity of osteoporosis requires versatile regulators such as miRNAs that can modulate multiple biological pathways. Recent studies have demonstrated the potential of miRNA-based therapy to restore bone homeostasis in osteoporotic models. However, further studies are needed to develop tissue-specific delivery systems and evaluate long-term safety to improve the therapeutic potential of miRNAs as new osteoporosis drugs.

## 1. Introduction

Bone is a complex and dynamic structure that functions as both a tissue and an organ system within higher vertebrates [1,2,3]. It is a highly specialized supporting framework of the body, characterized by its hardness, stiffness and ability to regenerate and repair itself [3]. In addition to protecting vital organs, bone tissue houses the bone marrow, which maintains hematopoietic stem cell homeostasis. It also serves as a reservoir of minerals that regulate calcium and phosphate levels in check, releases growth factors and cytokines, and contributes to the body’s acid–base balance [4]. The skeleton system is metabolically active and undergoes continuous turnover throughout life [5]. Once the skeleton has grown to maturity, it undergoes remodeling to continuously regenerate through periodic replacement of old bone with new bone at the same location. Bone remodeling involves the elimination of mineralized bone by osteoclasts (bone resorption) followed by the formation of bone matrix through osteoblasts (bone formation) becoming mineralized [6]. The process of bone remodeling is both systemic and local, and leads to complete regeneration of the adult skeleton every 10 years [7]. It is now widely accepted that bone remodeling aids in adjusting bone architecture to meet changing mechanical needs, repair fatigue damage in bone matrix, maintain plasma calcium homeostasis, and combat the effects of aging [5]. In bone remodeling, bone formation is tightly coupled to bone resorption, and occurs in a basic multicellular unit (BMU) of cells in the uninjured adult skeleton [8]. In BMUs, osteoclasts and osteoblasts maintain a well-orchestrated spatial and sequential association. Osteoclasts adhere to bone and subsequently eliminate it by acidification and proteolytic digestion. Osteoblasts then move in to cover the excavated area and initiate the process of new bone formation by secreting osteoid, which is finally mineralized into new bone [7,9]. A subset of osteoblasts further differentiate into osteocytes embedded in the bone matrix. Osteocytes sense mechanical stimuli and convert them into biochemical signals that regulate bone homeostasis [10,11]. The coordinated actions of bone resorption and bone formation contribute to maintain healthy bone in the adult population [7,9]. Any imbalance between these processes disrupts bone remodeling, resulting in the development of various bone diseases, including osteoporosis [4,9,12].

Osteoporosis is skeletal disorder characterized by the progressive loss of bone mineral density (BMD), deteriorated microarchitecture of the bone, and increased bone fragility. This causes pain, spinal deformity, and an increased risk of non-traumatic fractures of the hip, pelvis, spine, proximal humerus, and wrist, often requiring hospitalization [7,13,14]. Osteoporosis manifests at the cellular level as elevated bone resorption driven by excessive osteoclast formation and resorption activity, without sufficient compensatory osteoblast-mediated bone formation [12,15]. Osteoporosis is a major global public health concern that affects individuals of both sexes and all ethnicities, though it primarily affects elderly people and postmenopausal women [15]. The prevalence of osteoporosis is estimated at 18.3% worldwide with higher incidence in women (23.1%) than in men (11.7%), and the highest prevalence is observed in Africa (39.5%) [8,9,14,16]. Despite the large range of available treatments, osteoporosis remains substantially undertreated [12,16,17]. Drug treatments for osteoporosis are categorized as either anti-resorptive or anabolic. Anti-resorption drugs prevent bone loss by slowing down the pace at which osteoclasts reabsorb bone, but do not lead to an increase in bone mass. In contrast, anabolic drugs (anabolics) enhance bone formation by promoting osteoblastogenesis [12,17]. The most effective treatments have been found to be anti-resorptive medications such as bisphosphonates (e.g., alendronate, risedronate, ibandronate, zoledronic acid) [18,19], selective estrogen receptor modulators (SERMS; e.g., raloxifene), calcitonin (e.g., miacalcin and calcimar), and monoclonal antibodies (e.g., denosumab) [12,19,20]. Use of these medications has generated an abundance of evidence on their efficacy and safety, which has established risks and limitations for each. For example, bisphosphonates can cause uncommon, but severe complications such as atypical femur fractures and jaw osteonecrosis, and denosumab must be administered continuously to prevent rebound effects [19]. In recent decades, parathyroid hormone-related analogues (e.g., teriparatide and abaloparatide) [19,21] and anti-sclerostin agents (e.g., romosozumab) [20] have also been tested as anabolic therapies for osteoporosis, especially in individuals who are at high risk of fragility fractures [17]. Compared to anti-resorptive medicines, anabolic agents have demonstrated higher efficacy in increasing BMD and reducing the risk of fracture. However, their use is often limited by high costs, the need for daily injections, and the potential for causing adverse effects such as osteosarcoma [22]. In conclusion, current drugs for osteoporosis are limited by the risks of potential side effects and the requirement for frequent dosing in long-term treatments. Thus, there is an unmet need to develop novel therapeutic targets and strategies for restoring bone homeostasis in patients with osteoporosis while minimizing potential untoward side effects.

The emergence of “multi-omics” technologies (e.g., epigenomics, transcriptomics, proteomics, lipidomics, and metabolomics) has drastically advanced the study of osteoporosis in recent years by offering potent tools for examining the intricate molecular mechanisms of the condition. This enables us to better understand the molecular mechanisms controlling bone metabolism in osteoporotic conditions [15]. Among the key regulators of these pathways are microRNAs (miRNAs), which have emerged as important post-transcriptional modulators of gene expression involved in bone formation and resorption. Several miRNAs have been identified as potential novel osteoporosis therapy targets [23]. miRNAs are small, single-stranded, non-coding RNA molecules ranging from 19 to 24 nucleotides in length. miRNAs downregulate the expression of their target genes either through messenger RNA (mRNA) degradation or translational inhibition [24] by binding to complementary sequences in the 3′ untranslated region (3′UTR) of target mRNAs [25]. Although miRNAs preferentially interact with the 3′UTR, they have been reported to also bind to target sequences in the 5′UTR and coding region of mRNAs [26,27]. Despite making up only 1% to 5% of the human genome [28,29], miRNAs function as critical epigenetic regulators because they control the expression of up to 60% of protein-coding genes [29]. miRNAs have been found in cellular organelles such as nucleoli, processing bodies, and mitochondria, but most of them are located in the cytoplasm [30]. Additionally, miRNAs are encapsulated in extracellular vesicles such as exosomes to facilitate intercellular communication [31]. In recent years, numerous studies have shown that dysregulated expression of miRNAs contributes to the development of various human diseases. As a result, these tiny molecules are regarded as extremely promising targets for therapy as well as biomarkers for the diagnosis and prognosis of disease [32,33]. Due to their high sensitivity and specificity, miRNAs can be readily detected in tissue biopsies and body fluids. This makes them excellent non-invasive diagnostic biomarkers across a wide range of diseases [34,35]. Recent studies have reported critical roles of miRNAs in various biological processes, such as tissue development, homeostasis, repair, and disease, by regulating cell proliferation, apoptosis, cytoskeletal organization, signaling, metabolism, and differentiation [36]. In the context of bone health, many miRNAs have been linked to the progression of bone-related disorders [37,38,39]. These miRNAs govern bone remodeling by regulating the differentiation and function of osteoblasts and osteoclasts. For example, miR-20a, miR-26a, miR-29b, miR-125b, miR-133, miR-135, miR-196a, and miR-637 are mainly involved in regulating osteoblast differentiation, while miR-7b, miR-31, miR-26a, miR-145, miR-148a, miR-186, miR-340, and miR-365 control osteoclast differentiation [40]. Their dysregulation impedes bone metabolism, leading to osteoporosis via regulating the transforming growth factor-β (*TGF-β*)/bone morphogenetic protein *(BMP)* pathway, Dickkopf-1 *(DKK1)*, Runt-related transcription factor 2 *(RUNX2)*, the receptor activator of nuclear factor kappa B *(RANK)/RANK* ligand *(RANKL)*/osteoprotegerin *(OPG)* pathway, and *WNT* signaling [39,41,42,43]. Understanding these mechanisms offers promising opportunities to develop miRNA-based therapy for osteoporosis. Recent studies have shown that miRNAs not only play direct effects on osteoblasts and osteoclasts but also mediate the communication between these cells, which is essential for maintaining bone homeostasis. For instance, osteoclast-derived exosomal miR-214-3p is shown to inhibit the differentiation of neighboring osteoblasts by regulating ephrinA2/ephrin type A receptor 2 *(EphA2)* interactions, thereby establishing a direct link between osteoclast activity and reduced bone formation [44,45]. Moreover, it has been reported that mineralized osteoblasts release miR-503-3p-containing exosomes, which inhibit the differentiation of osteoclast progenitors by downregulating heparanase gene *(Hpse)* expression [46,47]. Finally, exosomal miR-218, derived from osteocytes, communicates with osteoblasts by downregulating sclerostin *(Sost)* expression, thereby promoting osteogenic differentiation [48].

This review highlights the current understanding of bone homeostasis and the molecular mechanisms underlying bone loss in osteoporosis. We explore the emerging roles of miRNAs in regulating osteoclast and osteoblast activity as well as their therapeutic potential. Additionally, we discuss current miRNA-based treatments for osteoporosis and address the challenges that must be overcome to enhance the efficacy of miRNA-mediated therapies.

## 2. MicroRNAs: Emerging Frontiers in Osteoporosis Research

Since the discovery of *lin-4* and *let-7* in *Caenorhabditis elegans*, miRNAs have been recognized to function as master regulators of post-transcriptional gene expression [49]. Approximately 50% of all identified miRNAs are intragenic and primarily derived from introns, although some are contained within the exons of protein-coding genes. Intergenic miRNAs are transcribed independently of a host gene and regulated by their own promoters [50,51], although miRNAs can occasionally be transcribed in clusters, which are single, lengthy transcripts that share seed areas and are regarded as a family [52]. Typically, miRNAs bind to complementary sequences within the 3′-UTR of their target mRNAs and downregulate gene expression by promoting mRNA degradation or inhibiting translation [23,53]. While the 3′-UTR is the most common binding region for miRNAs, some can target sequences with the 5′-UTR and coding regions of mRNAs [26,54]. The extent of base-pair complementarity at the miRNA binding sites determines the outcome of the interaction, and can lead to reduced mRNA stability and/or suppressed translation with sufficient binding affinity, thereby reducing mRNA and protein expression levels [53]. However, miRNAs can also activate gene expression under specific conditions [55]. Recent evidence indicates that miRNAs can be dynamically shuttled between subcellular compartments and can regulate transcription as well as translation [56].

### 2.1. Biogenesis of miRNAs

Endogenous miRNAs are generated through an intricate multi-step biogenesis process that is under tight spatial and temporal regulation: the canonical and non-canonical pathways (Figure 1). The predominant mechanism for processing miRNAs is the canonical biogenesis pathway, wherein a hairpin-like transcript known as primary miRNA (pri-miRNA) is generated from endogenous DNA loci by RNA polymerase II/III post- or co-transcriptionally [57]. This pri-miRNA undergoes stepwise processing by microprocessor (a heterodimer complex formed by Drosha and DiGeorge syndrome critical region 8 *(DGCR8)*) into a 70 to 120 nucleotides precursor miRNA (pre-miRNA) duplex structure [58,59,60]. The pre-miRNA is exported into the cytoplasm through exportin 5 (*XPO5*) [58] and then incorporated into the RNA-induced silencing complex (RISC) for a second cleavage event by Dicer, a ribonuclease that breaks down double-stranded RNA into smaller RNA species [61]. The RISC is a multi-protein complex made up of Argonaute RISC catalytic component 2 *(Ago2)*, astrocyte elevated gene-1, *R2D2* (a double-stranded RNA (dsRNA) binding protein with two dsRNA binding domains), fragile X intellectual disability 1, armitage-RNA helicase1, aubergine (another Argonaute family protein), vasa intronic gene, and Staphylococcal nuclease domain-containing protein 1 [62,63]. An adenosine triphosphate (ATP)-dependent process loads both strands of the mature miRNA duplex into the Argonaute proteins. After that, Ago2 removes one strand of the duplex [64]. The mature miRNA is then referred to as either the 5p strand (miRNA-5p) if it originates from the 5′-end of the pre-miRNA hairpin or the 3p strand (miRNA-3p) if it originates from the 3′-end, depending on the directionality of the miRNA’s strands [52,65]. The two mature miRNAs originating from two different strands of the same pre-miRNA exhibit different target specificities. The strand that remains incorporated in the RISC is the active strand, and it is designated as the guide strand, leading strand, or miR. The other strand gets degraded, and it is referred to as the passenger strand or miR* (* represents passenger strand also called the miRNA* strand of the miRNA duplex) [57,63]. The mature miRNA directs an Argonaute protein to silence target mRNAs in a seed sequence-dependent manner. Binding mostly happens at the 3′-UTR, but miRNAs can also bind to the 5′-UTR, coding regions, promoter sequences, or open reading frames of target genes. This controls their expression after transcription [26,52,54].

At present, several non-canonical miRNA synthesis routes have been identified that utilize various combinations of the canonical pathway’s components, primarily *Drosha*, *Dicer*, *XPO5*, and *Ago2.* Non-canonical miRNA biogenesis can generally be divided into two categories: Dicer-independent and Drosha/DGCR8-independent [52]. The first recognized non-canonical mechanism involves splicing-dependent “mirtrons”, which are short introns that are created after splicing and debranching and fold into pre-miRNA-like hairpins which bypass Drosha/DGCR8 processing, but still require Dicer for maturation [66]. Another example is 7-methylguanosine (m7G)-capped pre-miRNAs, which are directly exported to the cytoplasm via interacting with Exportin 1 without Drosha cleavage. While their subsequent maturation occurs via the canonical Dicer-dependent pathway, the m7G cap inhibits effective 5′-strand loading into Argonaute, resulting in a pronounced 3′-strand bias in miRNAs processed through non-canonical pathways, which has functional ramifications for the regulatory output and target specificity of miRNAs [52]. Conversely, specific miRNAs, including miR-451, are produced independently of Dicer processing and these pre-miRNAs are directly loaded into Argonaute after Drosha cleavage. Argonaute then uses its catalytic slicer activity to help them mature [67]. These different pathways show how flexible the miRNA biogenesis machinery is, since it can make functional miRNAs even when the usual parts are not there.

### 2.2. In Vivo Impact of miRNA Deletions or Mutations on Skeletal Development

MicroRNAs regulate numerous biological processes, including differentiation, proliferation, angiogenesis, apoptosis, migration, and immune regulation [57]. Their dysregulated expression is associated with various diseases, including cancer, diabetes, neurodegeneration, and cardiovascular disorders [34,35]. Recent in vivo studies have highlighted the essential role of miRNAs in regulating skeletal development, demonstrating their involvement in the pathophysiology of osteoporosis through the maintenance of bone homeostasis [23,37]. Mice lacking Dicer in skeletal progenitors (*Prx1-Cre*) develop short stature due to enhanced cell death during the initial stages of limb bud development [68]. Likewise, the loss of Dicer in osteoblasts *(Col1a1-Cre)* hindered extracellular matrix mineralization and diminished bone tissue [69]. Inducible deletion of Dicer in *Sp7^+^* osteoprogenitors at the post-natal stage diminished cortical bone mass in both juvenile and adult mice [70]. However, bone formation was increased by the deletion of *Dgcr8* in osteoprogenitors due to reduced miR-22 and increased *OCN* expression [71]. Other examples of how miRNA deletions or mutations affect skeletal development are summarized in Table 1.

## 3. Impact of Altered miRNA Expression on Bone Cell Function and Osteoporosis

MicroRNA studies have demonstrated that altered miRNA expression profiles affect the differentiation and function of osteoclasts, osteoblasts, and osteocytes, disrupting bone remodeling [23]. Numerous miRNAs have been reported to regulate key factors and signaling pathways, including *TGF-β/BMP*, *WNT/β-catenin*, *NOTCH*, and *RANKL/OPG* signaling pathways, whose dysregulation is associated with osteoporosis. The miRNAs promoting osteoblast differentiation are miR-10b [89], miR-19a-3p [90], miR-26b [91], miR-29a [92], miR-130a [93], and miR-374b [94], which target *SMAD2*, *HDAC4*, *GSK3β*, *SMAD6*, *DKK1*, *SMURF2*, and *PTEN*, respectively. Conversely, miR-9-5p [95], miR-16-2-3p [96], miR-23a [97], miR-27b-3p [98], miR-93-5p [99], miR-125b [100], miR-137-3p [101], miR-451a [102], and miR-1187 [103] have been reported to inhibit osteoblast differentiation by targeting *WNT3A*, *WNT5A*, *BMPR1B*, *SP7/OSX*, *BMP2*, *BMPR1B*, *RUNX2*, *BMP6*, and *ARHGEF-9*, respectively.

As miRNAs regulating osteoclast differentiation, miR-144-3p and miR-133a modulate the *RANKL-RANK-OPG* ligand–receptor signaling axis [104]; miR-503 directly targets *RANK* [105]; miR-124 inhibits nuclear factor of activated T cells 1 (*NFATc1*) [106]; miR-338-3p promotes osteoclast differentiation by targeting the *MafB* transcription factor [107]; miR-214 increases osteoclastogenesis through activation of the *PTEN/PI3K/AKT* signaling pathway [108]. Recent studies have identified the distinct roles of miRNAs in osteoblast and osteoclast functions, which contribute to osteoporosis. Nevertheless, elucidating the precise relationships between miRNAs and osteoporosis still remains in its early stages, requiring additional research to discover reliable biomarkers and therapeutic targets.

### 3.1. Osteoblast-Regulating miRNAs

Osteoblasts are important for keeping bone mass stable while bones are growing and changing shape. To maintain bone homeostasis, the interaction between osteoclasts and osteoblasts is tightly regulated by both systemic and local factors [109]. Both genetic and epigenetic factors tightly regulate the balance between resorption and formation. miRNAs strongly impact osteoblast differentiation at the stages of lineage commitment and maturation by regulating gene expression within cells as well as by controlling the transduction of osteogenic signaling pathways [110] (Figure 2). Therapeutic strategies targeting osteoblast-specific miRNAs that promote bone formation have emerged as a promising approach for treating osteoporosis (Table 2). The transcription factors, including *RUNX2*, *OSX/SP7*, and *DLX5*, and *WNT*, *NOTCH*, *TGF-β*, and *PI3K/AKT* signaling pathways have been known as key players of osteoblastogenesis [111]. The expression and activities of these transcriptional factors and signaling pathways are tightly regulated by epigenetic mechanisms, including miRNAs. For instance, miR-433-3p promotes bone formation by blocking *DKK1*, an WNT antagonist [112], whereas upregulation of *WNT5A*-targeting *miR-16-2** disrupts bone metabolism by reducing *RUNX2* activity [96]. Furthermore, miR-124 has gained attention for its inhibitory role in osteoblast differentiation. Overexpression of miR-124 suppresses key osteogenic transcription factors such as *Dlx5*, *Dlx3*, and *Dlx2*, while its inhibition enhances osteoblast marker expression, *ALP* activity, and matrix mineralization. Moreover, miR-124 inhibition significantly increased ectopic bone formation in vivo, suggesting that miR-124 acts as a negative regulator of osteogenesis by targeting the *Dlx* gene family [113]. In addition to miRNAs that regulate osteoblast differentiation through classical signaling pathways, mechano-sensitive miRNAs have emerged as critical modulators of bone formation in response to mechanical stimuli. The first mechano-sensitive miRNA identified was miR-103a, which inhibits osteoblast development by targeting *Runx2.* Therapeutic inhibition of *miR-103a* could be an effective anabolic treatment for skeletal problems caused by pathological mechanical stress [114]. Another novel mechano-sensitive miRNA is miR-33-5p, which is a potential target for treating pathological bone loss as it can stimulate osteoblast development and regulates differentiation triggered by modifications in the mechanical environment [115]. The details of miRNAs that have been reported to regulate osteoblast function and differentiation are presented in Table 2.

### 3.2. Osteoclast-Regulating miRNAs

The bone marrow monocytes (BMMs) give rise to multinucleated, bone-resorbing cells called osteoclasts [155]. Maintaining balance between osteoclastogenesis and osteogenesis is essential for bone integrity, and its disruption can lead to bone fragility. The involvement of miRNAs specifically in osteoclasts has recently gained significant attention, driven by therapeutic target to attenuate excessive bone resorption in pathological conditions such as age, estrogen deficiency, cancer metastases, or glucocorticoid use [109]. miR-182, miR-31, miR-25-3p, miR-21-5p, miR-214-3p, miR-125a-5p, miR-125a-3p, miR-155, and miR-221-5p play crucial roles in regulating osteoclast differentiation and activity, making them key players in developing osteoporosis (Table 3; Figure 2). For instance, deletion of miR-182 in myeloid cells protects against excessive osteoclastogenesis and bone resorption in a mouse model of ovariectomy (OVX)-induced osteoporosis. This protective effect is mediated via its regulation of dsRNA-dependent protein kinase *(PKR)*, which controls the autocrine feedback loop that is mediated by *IFN-β.* These findings demonstrated the miR-182*-PKR-IFN-β* axis as a novel inhibitory pathway of osteoclastogenesis [78]. Additionally, high-throughput miRNA sequencing has identified miR-182 as a central regulator of *TNF-α*-induced inflammatory osteoclastogenesis, a process normally restrained by the Notch pathway mediator *RBP-J*. In *RBP J* deficient cells, suppression of *miR-182* markedly reduces *TNFα*-driven osteoclast formation. Moreover, miR-182 directly targets forkhead box, class O *(FoxO)* 3 and mastermind-like 1 *(Maml1)*, both of which inhibit osteoclast development. Thus, targeting of *RBP-J*-miR-182*-FoxO3/Maml 1* signaling axis could be a promising therapeutic approach to decrease inflammatory osteoclastogenesis and bone resorption [156]. miR-31 has been reported as a crucial regulator of osteoclastogenesis and cytoskeleton organization. miR-31 targets *Ras* homolog family member A *(RhoA)*, a molecular switch that controls cytoskeletal organization. Its inhibition reduced osteoclast development, bone resorption, and actin ring formation [157].

miR-125a expression is downregulated during osteoclast differentiation. Its overexpression inhibits the activity of tumor necrosis factor receptor-associated factor 6 *(TRAF6)*, which is an essential activator of the *RANKL/RANK/NFATc1* signaling pathway and may further block osteoclast formation. Its potential as a potent anti-resorption agent has been highlighted in the context of osteoporosis therapy [158]. Huang et al. identified miR-21-5p as an osteoclastogenesis regulator, demonstrating that its overexpression reduces osteoclast differentiation by targeting *SKP2*, while its downregulation reverses this effect [159]. These findings indicate that miR-21-5p may be another potential therapeutic target for osteoporosis. Osteoclast-derived exosomes carrying long non-coding RNA (lncRNA) AW011738 suppress osteogenesis in OVX mice while inhibiting the osteogenic differentiation of MC3T3-E1 cells by exerting a controlling effect on the *lncRNA AW011738/*miR-24-2-5p*/TREM1* axis [160]. Thus, miR-24-2-5p-mediated regulation of lncRNA *AW011738* may emerge as a new therapeutic target for osteoporosis in postmenopausal women. Elevated expression of osteoclastic miR-214-3p increases serum exosomal miR-214-3p levels and suppresses bone formation in elderly women with fractures and in OVX mice [44]. Similarly, osteoclast-specific miR-214-3p knock-in mice show elevated serum exosomal miR-214-3p and reduced bone formation, and these effects can be reversed by osteoclast-targeted antagomiR-214-3p therapy. Moreover, inhibition of miR-214-3p in osteoclasts enhances bone formation in aging OVX mice. Collectively, these findings indicate that osteoclast-targeted inhibition of miR-214-3p may represent a promising therapeutic approach for osteoporosis [44].

In recent years, our understanding of the roles of miRNAs in osteoclast differentiation and activity in the context of osteoporosis therapy has expanded significantly. Despite extensive studies in vitro or ex vivo, in vivo and clinical research are required to validate the phycological roles of miRNAs in bone homeostasis. Advanced miRNA biology should enable scientists to fully harness their therapeutic potential as disease-modifying agents in osteoporosis (Table 3).
cells-14-01905-t003_Table 3Table 3Key miRNAs involved in osteoclastogenesis and their therapeutic implications in osteoporosis.miRNATarget*(s)*Therapeutic OutcomeReferencemiR-21*PDCD4*, *FasL*Inhibition of miR-21 increases bone mass with reduced osteoclast number and resorption activity.[161,162]miR-29 family*NFIA*, *CDC42*, *SRGAP2*, *GPR85*, *CD93*Overexpression of miR-29 promotes osteoclast precursor commitment and migration; inhibition promotes macrophage differentiation.[163]miR-31*RhoA*Overexpression of miR-31 promotes actin ring formation and bone resorption.[157]miR-34a*TGIF2*Overexpression of miR-34a increases bone mass by inhibiting osteoclastogenesis.[164]miR-34c*LGR4*Overexpression of miR-34c sustains osteoclast survival by repressing *LGR4*-mediated apoptosis.[165]miR-125a*TRAF6*Overexpression of miR-125a suppresses osteoclastogenesis through the *TRAF6/NFATc1* loop.[166]miR-128*Sirt1*Inhibition of miR-128 increases bone volume by targeting *Sirt1* and suppressing *NF-κB* pathway, thereby reducing osteoclast differentiation and bone resorption.[167]miR-141*EPHA2*, *CALCR*Overexpression of miR-141 inhibits osteoclast differentiation and bone resorption.[168]miR-144-3p*RANK*Overexpression of miR-144-3p inhibits osteoclast precursor survival and proliferation.[169]miR-145*Smad3*Overexpression of miR-145 reduces osteoclast differentiation and bone resorption[170]miR-146a*TRAF6*Overexpression of miR-146a inhibits osteoclastogenesis and reduces arthritis-induced bone resorption.[171]miR-148a*MAFB*Overexpression of miR-148a promotes osteoclast differentiation; inhibition protects against OVX-induced bone loss.[172]miR-155*SOCS1*, *MITF*, *TAB2*Overexpression of miR-155 suppresses osteoclastogenesis under *IFNβ/TGFβ* but promotes activity under LPS stimulation.[173,174]miR-183*HMOX1*Overexpression of miR-183 promotes osteoclast differentiation through ROS regulation.[175]miR-186*CTSK*Overexpression of miR-186 reduces osteoclast survival.[176]miR-193-3p*NFATC1*, *CTSK*, *ACP5*, *CAR2*Overexpression of miR-193-3p reduces osteoclastogenesis and bone resorption.[177]miR-199a-5p*MAFB*Overexpression of miR-199a-5p promotes osteoclast formation via *NFATc1* amplification.[178]miR-214*PTEN*, *ATF4*Overexpression of miR-214 enhances osteoclast differentiation and bone resorption; exosomal miR-214 from osteoclasts inhibits osteoblast bone formation.[44]miR-218*TNFRSF1A*, *NF-κB pathway*Overexpression of miR-218 inhibits osteoclastogenesis and protects against OVX-induced bone loss.[179,180]miR-223*NFIA*Overexpression of miR-223 suppresses osteoclast differentiation by targeting *NFIA* and downregulating *NFATc1* signaling.[181,182]miR-301b*CYLD*Inhibition of miR-301b reduces osteoclastogenesis and protects from OVX-induced bone loss.[183]miR-340*MITF*Overexpression of *miR-340* inhibits osteoclast differentiation.[184]miR-365*MMP9*Overexpression of miR-365 reduces bone resorption.[185]miR-100-5p*FGF21*Overexpression of miR-100-5p suppresses bone resorption.[186]miR-539-3p*Apoptotic regulators*Inhibition of miR-539 prevents osteoclast differentiation and bone resorption.[139]miR-182*PKR*Myeloid-specific deletion protects against excessive osteoclastogenesis and bone resorption; pharmacological treatment with antagomiRs completely suppresses pathologic bone erosion.[78]*FoxO3 and Maml1*Inhibition of miR-182 with antagomiRs suppresses *TNF-α*-induced osteoclast formation and bone degradation[156]miR-31*Rho A*Inhibition of miR-31 by specific antagomiRs suppresses the *RANKL*-induced osteoclast development, bone resorption and actin ring formation.[157]miR-25-3p*NFIX*Overexpression of miR-25-3p inhibits *NFIX* expression, suppressing osteoclast proliferation.[187]miR-21-5p*SKP2*Overexpression of miR-21-5p in vitro reduces osteoclast differentiation and activity; pre-miR-21-5p treatment in OVX mice inhibits bone resorption and maintains bone cortex and trabecular structure.[159]miR-214-3p*PTEN*Osteoclast-targeted antagomiM-214-3p therapy reduces the serum exosomal miR-214-3p and enhances bone formation. [44]miR-125a-3p*TRAF6*Overexpression of miR-125-3p inhibits the osteoclast differentiation and osteoclast-driven bone resorption.[158]miR-155*TAB2*Inhibition of miR-155 reduces inflammatory osteoclastogenesis through autophagy suppression.[174]*LEPR*Inhibition of miR-155 reduces osteoclast activation and bone resorption.[188]miR-221-5p*Smad3*Overexpression of miR-221-5p alleviates postmenopausal osteoporosis (PMO) through suppressing osteoclastogenesis.[189]


### 3.3. miRNAs Regulating Both Osteoclasts and Osteoblasts

miRNAs are small disease modifying molecules that have emerged as critical regulators of both osteoblast and osteoclast lineages. These post transcriptional messengers either promote or suppress the function and differentiation of either osteoblasts or osteoclasts, affecting bone remodeling and homeostasis. Studies have shown that several miRNAs control the differentiation, activity, and survival of both osteoblasts and osteoclasts and provide the basis for maintaining the equilibrium between bone formation and resorption (Figure 2; Table 4). miR-214 not only inhibits osteoblast function by targeting activating transcription factor 4 (*ATF4*) [81], but it also promotes osteoclastogenesis by activating the *PI3K* pathway and inhibiting *PTEN* [108]. In addition, exosomal miR-214-3p derived from osteoclasts acts as a coupling factor in inhibiting neighboring osteoblasts [44]. Sun et al. demonstrated a new mechanism of osteoclast-osteoblast crosstalk in which miR-214-carrying exosomes secreted from osteoclasts inhibit osteoblast function through interactions with *ephrinA2/EphA2* proteins [45]. miR-324 has been also reported to regulate the function of osteoblasts and osteoclasts by controlling cellular biochemical processes in the latter. miR-324 inhibits osteogenesis through targeting of *Runx2*, a master transcription factor of osteoblast development. Elevated levels of *Runx2* in miR-324-deficient mice promotes bone formation via augmented osteoblast development while suppressing adipogenesis. On the other hand, miR-324 targets *Pin1* in osteoclasts, enhancing osteoclast fusion. Deletion of miR-324 reduces *Pin1* expression, which inhibits osteoclastogenesis and bone resorption while promoting osteoblast differentiation. These dual activities of miR-324 indicate its importance in maintaining bone homeostasis and hence makes it a promising therapeutic target for bone-related disorders such as osteoporosis [190]. In contrast, miR-34a-5p inhibits osteoclast differentiation by targeting *RANKL* and *HIF1A* while promoting osteoblast differentiation by inhibiting the *Notch1* and *DKK1* signaling pathway [191,192]. Thus, further study of the dual functional miRNAs as regulators of both osteoblasts and osteoclasts is crucial for improving our understanding of how the delicate balance between bone formation and reabsorption is maintained (Table 4). Any disruption in this regulation could lead to bone disorders, and so miRNAs could be potential targets for future therapeutic interventions to re-balance this process in disease states.

## 4. Current microRNA-Based Therapeutics and Diagnostics in Osteoporosis

### 4.1. Diagnostic Applications of miRNAs in Osteoporosis

miRNAs have been developing as potential therapeutics in the pharmaceutical sector. Due to their biocompatibility, evolutionary conservation, easy modification, pleiotropy, and high efficacy, miRNAs hold the potential to revolutionize the diagnosis and treatment of various complex diseases, including bone-associated disorders [195]. Furthermore, the stability of miRNAs in peripheral fluids like blood, urine, and cerebrospinal fluid makes them effective non-invasive biomarkers for determining disease diagnosis and prognosis. Their presence in extracellular vesicles, protein complexes, or as free-circulating molecules protects them from RNase degradation, allowing them to be reliably detected using liquid biopsy procedures [195]. miRNA-based therapies can be used to reduce overexpressed miRNA (loss-of-function) or restore the expression of downregulated or non-functional miRNA (gain-of-function).

Techniques, including miRNA sponges, miRNA masks, locked nucleic acid (LNA) inhibitors, and anti-miRNA oligonucleotides (anti-miRs), have been developed to downregulate miRNA expression. These molecules bind specifically to the target miRNA, preventing interaction with target mRNAs. The resulting miRNA-inhibitor duplexes are subsequently degraded by RNase H. Strategies to restore miRNA function are being investigated using miRNA mimic and viral vector-mediated delivery approaches [196,197].

Biomarkers are measurable indicators of a biological state or condition that reflect alterations within a system, enabling the detection of disease onset and progression with high sensitivity and specificity [198]. Recent advances in biomedical research have identified several miRNAs as promising biomarkers for osteoporosis management. For example, miR-148a-3p, which targets *MAFB*, *PPAR*, and *WNT1* transcripts, has demonstrated high potential in discriminating postmenopausal osteoporosis (PMO) patients from healthy controls [199]. Likewise, the OsteomiR™ panel is a potential diagnostic reagent for patients with PMO and at risk for fractures [200]. The details of potential miRNA biomarkers in PMO are presented in Table 5.

### 4.2. Therapeutic Applications of miRNAs in Osteoporosis

Therapeutic miRNAs have emerged as an important class of biopharmaceutical drug that is in the commercial space as future medicines. One of the major advantages of miRNAs and miRNAs-targeting oligonucleotides as therapeutic agents over traditional small-molecule drugs is that their oligonucleotides can be chemically altered to improve their pharmacokinetic/pharmacodynamic profiles, and they have the capability to target several genes simultaneously. It is important to note that sensitivity of oligonucleotides to degradation by serum nucleases encouraged the hunt for chemical modifications of oligonucleotides so that there is increase in the stability and efficacy of these oligonucleotides in vitro and in vivo studies [218,219]. Several chemical modifications including 2′-O-Methylation (2′-O-Me), 2′-fluoro modification (2′-F) [220], LNA technology [221], phosphorothioate (PS) alterations [222], peptide nucleic acids (PNA), and pH low-insertion peptide (pHLIP) modifications [197] have been introduced that enhance the stability and efficacy of miRNA-based therapeutics [220,223]. Chemical modifications, including 2′-O-Me and 2′-F, increase miRNA stability and binding affinity through structural alterations of the ribose sugar [224]. LNA technology enhances the target specificity and resistance of miRNAs to degradation, making them valuable tools for therapeutic applications [225]. Furthermore, PS modifications introduce sulfur into the phosphate backbone, thereby increasing the nuclease resistance and prolonging the half-life of miRNA molecules. This technique is employed for the selective delivery of synthetic anti-miR-21 (RG-012/lademirsen/SAR339375) into the kidney of patients in a clinical trial of Alport syndrome (NCT03373786, NCT02855268) [226]. PNA is a nucleic acid analogue that has the sugar-phosphate backbone replaced with a peptide-like structure, improving binding strength and biological stability. This modification allows PNAs to form very stable duplexes with the target miRNAs to block their interactions with target mRNAs [197]. Together, these changes address key challenges in miRNA-based therapy, including degradation, off-target effects, and bioavailability.

A major challenge in miRNA therapeutics is to develop targeted delivery of miRNA in vivo. For instance, free RNA molecules are highly susceptible to nuclease degradation and excretion by organs like the kidneys and liver. Other biological barriers, including the cell membrane, blood–brain barrier, and connective tissues, also hinder the in vivo delivery of miRNAs. Additionally, synthetic miRNAs can be recognized as foreign antigens, which can trigger immune responses. Finally, miRNAs may interact non-specifically with non-complementary or partially complementary mRNAs in targeted cells, or be trapped within endosomes [227]. To overcome these challenges, conjugation of ligands, antibodies, and nanoparticles, has been shown to improve the tissue-targeting specificity of miRNA and minimize off-target effects. Delivery systems for miRNA therapeutics are broadly studied using virus- and non-virus-based approaches. Virus-based systems utilize lentiviruses, bacterial nanocells, and adeno-associated viruses. Extracellular vesicles, also known as exosomes, liposomes (such as cationic lipid-based ligand-targeted nanoparticles), and natural and synthetic polymer-based nanoparticles (such coupling with polyethylene glycol) are non-virus-based mechanisms of delivery. Moreover, gold nanoparticles, carbon nanotubes, quantum dots, core–shell magnetic nanoparticles, and mesoporous silica nanoparticles are alternative nanocarriers used as drug delivery vehicles to maximize the efficiency and precise distribution of miRNA-based therapeutics [223].

As described above, miRNAs are highly susceptible to degradation by ribonucleases in vivo. Additionally, hydrophobic nature of the cell membrane hampers miRNA transport across the plasma membrane, thereby limiting their cellular uptake. Therefore, it is important to design appropriate carriers and adopt optimal delivery strategies for therapeutic miRNAs to target specific organs, thereby maximizing their therapeutic potential while minimizing off-target adverse effects [228]. Due to these challenges, local delivery of miRNA is more effective than systemic delivery, as it allows for targeted and sustained release at the injected site, minimizing off-target effects and systemic degradation However, local delivery to bone tissue is limited by its inability to reach deeper regions of the bone, and uneven local release can cause variable therapeutic outcomes. Most importantly, this delivery approach is unable to treat systemic or multifocal bone diseases [229]. A brief overview of the advantages and disadvantages of both virus-based and non-virus-based miRNA delivery systems is provided in Table 6.

Since the discovery of an association between miRNAs and human diseases in 2002, there has been exceptionally growing interest in exploring the potential of miRNAs as a novel class of therapeutics. Furthermore, the global success of *SARS-CoV-2* mRNA vaccines in combating the Coronavirus Disease 2019 (COVID-19) pandemic has sparked investigation into RNA-based immunotherapies, including miRNA-based approaches to treat complex diseases [223]. Nevertheless, the field of miRNA-based therapeutics remains in its budding phase, with only a few reaching the clinical development stage and none having yet successfully progressed to Phase III clinical trials or been approved by the U.S. Food and Drug Administration (FDA). Additionally, safety is a concern in moving these technologies forward in the clinic, as most of the clinical trials have been terminated due to off-target toxicities.

In 2013, MRX34, a miR-34a mimic, emerged as the first miRNA-based cancer therapeutic to move to a Phase I clinical trial. Endogenous miR-34a regulates the expression of *p53* as a tumor suppressor and is downregulated in numerous malignancies. However, clinical trials of MRX34 had to be terminated due to serious immune-associated toxicities and patient fatalities [227,255]. Remlarsen (also known as MRG201), a miRNA-29 mimic, was developed by Viridian Therapeutics (formerly known as miRagen Therapeutics). Despite initial promising results of miR-29b mimic against skin fibrosis, participants were reported to experience severe immune responses, leading to its discontinuation after the phase II trial [256]. Likewise, Miravirsen an LNA-modified antisense phosphorothioate inhibitor of miR-122, was the first miRNA therapy to enter phase II clinical trials primarily for evaluating its safety and efficacy against HCV (hepatitis C virus) infection. Similarly, RG-101, a GalNAc-conjugated inhibitor of liver-specific miR-122 developed by Regulus Therapeutics, also advanced into clinical testing for HCV, however, due to safety issues, the Phase II trials for both these drugs were discontinued [257,258,259]. A Phase II trial of an anti-miR targeting miR-21 by Regulus Therapeutics for the treatment of Alport syndrome was also discontinued early in 2022 due to safety concerns and a lack of efficacy [260].

Numerous clinical studies have demonstrated a strong association between osteoporosis and miRNAs. To date, eleven clinical trials have been registered in the ClinicalTrials.gov database. These clinical trials explore the role of molecular biomarkers, mainly miRNAs and lncRNAs, in osteoporosis, bone metabolism, and related conditions such as sarcopenia, diabetes mellitus obesity, parathyroid neoplasms, primary hyperparathyroidism, and PMO. The majority of these clinical trials focus on the finding of miRNAs as potential biomarkers (NCT03931109, NCT02128009, NCT02705040) or how they respond to different treatments, such as bisphosphonates, denosumab, teriparatide, and new dietary supplements (NCT03472846, NCT05328154, NCT05421819). Some studies have revealed the relationship between bone health and other systems, including cardiovascular function, sarcopenia, and pain (NCT05228262), metabolic disorders (NCT05673837), and intermittent fasting with exercise in postmenopausal women with obesity (NCT05912309). These trials commonly emphasize the use of molecular and epigenetic analyses to improve the diagnosis, treatment, and prevention strategies for osteoporosis and bone-related disorders. A brief overview of these trials is described in Table 7.

Given that each miRNA can control the expression of numerous genes, one of the primary issues with miRNA-based therapies is the emergence of off-target effects and related toxicities. While these therapies demonstrate great potential as future treatments, their broad effects could harm non-targeted tissues in the body [261]. One prominent example is MRX34, a liposomal nanoparticle (NOV40) encapsulating a miR-34a mimic that was evaluated in a Phase I trial for advanced solid tumors (NCT01829971). Despite displaying efficacy, three individuals had prolonged partial responses, and 14 patients had stable diseases. This trial was discontinued due to the death of four patients caused by immune-related adverse effects [255,261,262]. Moreover, pre-clinical studies have shown that miR-34a is not only taken up by the tumor tissue, but also accumulates in the spleen and bone marrow, key organs for immune cell production [263,264]. Off-target toxicities associated with miR-29, miR-122, and miR-21 have been reported, leading to the termination of several clinical trials at Phase I or Phase II [227,256,258,259,260]. Although these trials were discontinued, they provide valuable insights on the development of more precise delivery systems and optimized therapeutic payloads to reduce off-target toxicities and improve targeting strategies, potentially bringing miRNA-based therapies closer to clinical success.

## 5. Conclusions and Future Directions

MicroRNAs are key regulators of osteoblast and osteoclast function, orchestrating bone formation and reabsorption. In addition to their direct roles in osteoblasts and osteoclasts, miRNAs are also critical for the communication between these cells, required for maintaining bone remodeling balance. For example, exosomal miRNAs, including miR-214-3p, miR-503-3p, and miR-218, play roles in mediating the interaction between osteoblasts and osteoclasts. Because of these dual abilities, miRNAs represent promising therapeutic candidates for targeting both osteoblasts and osteoclasts to restore bone homeostasis. Recent studies have shown that miRNAs play important roles in osteoporosis by efficiently modulating complex signaling networks. MiRNA-based therapeutics that modulate bone remodeling activity (e.g., miRNA mimics to stimulate osteogenic microRNA activity and antagomiRs to inhibit bone-resorptive miRNAs) are promising therapeutic strategies for osteoporosis. There has been a growing interest in dual-targeting miRNA-based therapeutic approaches and their potential to regulate osteoblast and osteoclast activity simultaneously. This approach would further facilitate a comprehensive, individualized treatment modality for the multifactorial nature of osteoporosis. Moreover, miRNAs hold great promise as non-invasive biomarkers for the progression of osteoporosis. Circulating miRNAs in peripheral bodily fluids enable early diagnosis and prognosis and personalized therapy strategies. Although numerous miRNAs have been identified as potential regulators of bone remodeling for the treatment of osteoporosis, their translation from in vitro, in vivo, and pre-clinical studies to clinical trials remains in its early stages.

To date, no miRNA-based therapy (such as miRNA mimics or antagomiRs) has been approved for clinical use. Several factors contribute to this challenge, including obstacles related to delivery, safety, regulatory considerations, and off-target effects. Strategies to resolve such challenges include the optimization of delivery systems, stabilization of product, dosage optimization, and the reduction in causes of off-target effects. The development and application of miRNAs as diagnostic and therapeutic targets for osteoporosis require coordinated efforts from researchers, clinicians, and industry stakeholders to drive progress at the forefront of the field. These initiatives have been integrating state-of-the-art technologies including AI, machine learning, single-cell miRNA profiling, and multi-omics data approaches into miRNA research that facilitates a new transformative period of innovation in diagnostics and therapeutics. AI and machine learning algorithms are employed to analyze large volumes of data, enabling the early identification of new diagnostics, optimization of therapeutic strategies, and prediction of treatment outcomes for miRNAs. Single-cell RNA sequencing can measure miRNA expression at the individual cell level, providing a more precise assessment of cellular heterogeneity and potentially improving the accuracy of osteoporosis diagnosis and treatment.

## Figures and Tables

**Figure 1 cells-14-01905-f001:**
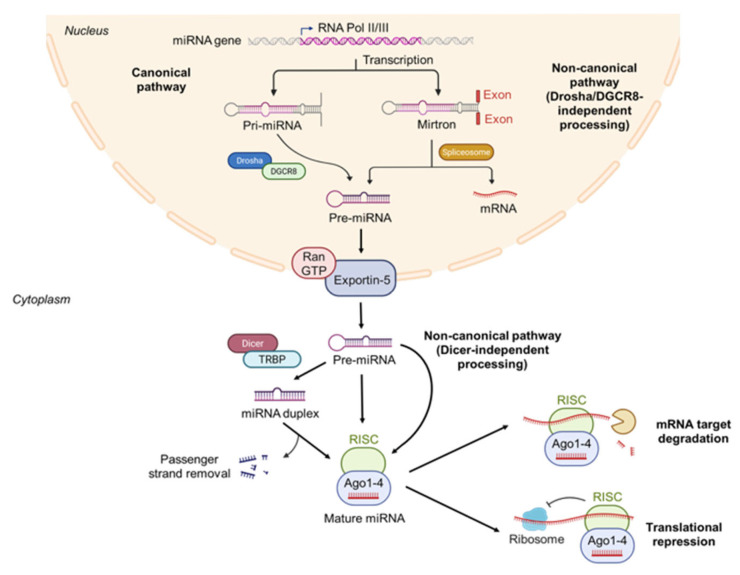
Canonical and non-canonical miRNA biogenesis pathways. In the canonical pathway of miRNA biogenesis, primary miRNA is processed in a stepwise manner by Drosha/DGCR8 and Dicer, with Exportin-5 mediating cytoplasmic export, and Argonaute loading the guide strand into RISC for mRNA silencing via degradation or translational repression. In the non-canonical Dicer-independent route, Ago2 directly cleaves pre-miRNA, bypassing Drosha or Dicer steps, to generate a mature RISC complex that mediates gene repression. This figure was created using BioRender.

**Figure 2 cells-14-01905-f002:**
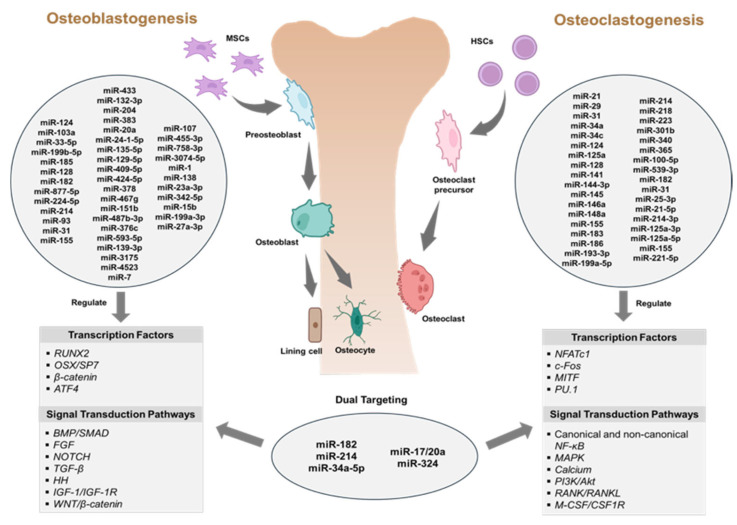
Illustration of the molecular regulation of osteoblastogenesis and osteoclastogenesis, highlighting transcription factors, signaling pathways and miRNAs. MSCs differentiate into osteoblasts through a cascade of molecular events regulated by key transcription factors such as *RUNX2*, *OSX/SP7*, *β-catenin* and *ATF4*, activated through signaling pathways including *BMP/SMAD*, *FGF*, *NOTCH*, *TGF-β*, *HH*, *IGF-1/IGF-1* receptor *(IGF-1R)* and *WNT/β-catenin*. In parallel, hematopoietic stem cells (HSCs) give rise to osteoclasts, guided by transcription factors such as *NFATc1*, *c-Fos*, microphthalmia-associated transcription factor *(MITF)* and purine-rich box-1 *(PU.1)*, under the influence of the *NF-κB*, *MAPK*, *calcium*, *PI3K/Akt*, *RANKL-RANK*, and *M-CSF/CSF1R* signaling pathways. miRNAs regulate these differentiation pathways in different ways. For instance, miR-124, miR-103a, miR-33-5p, and miR-199b-5p regulate osteoblastogenesis, while miR-21, miR-29, and miR-3 influence osteoclastogenesis. Additionally, multiple miRNAs, including miR-182 and miR-214, miR-34a-5p, miR-17/20a, and miR-324, which operate as dual regulators, target both osteoblast and osteoclast differentiation pathways.

**Table 1 cells-14-01905-t001:** Effects of miRNA deletions or mutations on skeletal development.

Finding	Affected miRNA/Gene	Effect on Skeletal Development	Reference
Mutation in 3′UTR of *HDAC6*	Disrupted miR-433 binding site	X-linked chondrodysplasia (dominant).	[72]
miR17-92 cluster deletion or duplication	Loss or gain of miRNA dosage	Microcephaly, abnormal facial features, short stature, and digit abnormalities.	[73]
Deletion of miR17-92 in *Col1α^2+^* cells (mice)	Reduced miRNA cluster activity	Smaller bones and decreased periosteal bone formation under mechanical loading.	[74]
Mutation in the gene encoding miR-140	Loss of miR-140 function	Short stature, brachydactyly, degeneration of intervertebral discs and delayed epiphyseal ossification of the hip and knee.	[75]
Deletion of *miR-140* (mice)	Loss of miR-140 dosage	Short limbs.	[76]
Deletion of miR-148a (mice)	Loss of miR-148a dosage	Increased bone mass.	[77]
Deletion of miR-182 in osteoclast precursors (mice)	Loss of function in osteoclasts	Protects against ovariectomy-induced osteoporosis and inflammatory arthritis.	[78]
Overexpression of *miR-29* in *Prx1^+^* MSCs (mice)	Gain of miR-29 dosage	Enhances bone mass, accelerates calvarial defect regeneration and reduces lipopolysaccharide (LPS)-induced organ injuries and mortality.	[79]
miR-455-null mouse	Loss of function	Dysregulation of bone turnover.	[80]
Overexpression of miR-214 in osteoblasts (mice)	Gain of function	Inhibitory role in bone formation.	[81]
Transgenic overexpression of miR-335-5p in *OSX*^+^ cells	Gain of miR-333-5p dosage	Enhances osteogenic differentiation and bone formation.	[82]
Transgenic overexpression of miR-188 in *OSX^+^* osteoprogenitors	Gain of miR-188 dosage	Greater age-associated bone loss and fat accumulation in bone marrow.	[83]
Transgenic mice overexpressing miR-29a driven by phosphoglycerate kinase *(PGK)* promoter	Gain of miR-29a dosage	Improved bone structure with reduced sensitivity to glucocorticoid-induced mineral and osteogenic loss.	[84]
Deletion of miR-34b/c in *Col1a1^+^* cells (mice)	Loss of function	Increased bone mass embryonically and postnatally.	[85]
Overexpression of miR-34c in *Col1a1^+^* cells (mice)	Gain of miR-34c dosage	Reduced bone mass.	[85]
Conditional overexpression of miR-206 in *Col1a1^+^* cells (mice)	Gain of miR-206 dosage	Low bone mass.	[86]
Global deletion of miR-181 family (mice)	Loss of function	Smaller body size; impaired growth and bone development.	[87,88]

**Table 2 cells-14-01905-t002:** Key miRNAs involved in osteoblastogenesis and their therapeutic implications in osteoporosis.

miRNA	Target(s)	Therapeutic Outcome	Reference
miRNA-124	*Dlx3*, *Dlx5* and *Dlx2*	Inhibition of miR-124 enhances osteogenic differentiation and ectopic bone formation.	[113]
miR-103a	*Runx2*	Inhibition of miR-103a restores bone formation under mechanical unloading conditions.	[114]
miR-33-5p	*HMGA2*	Overexpression of miR-33-5p promotes osteoblast differentiation and bone formation under mechanical stress.	[115]
miR-199b-5p	*GSK-3* *β*	Overexpression of miR-199b-5p enhances *ALP* and *Runx2* expression and *ALP* activity, thereby enhancing osteoblast differentiation.	[116]
miR-185	*PTH*	Inhibition of miR-185 promotes osteoblast proliferation and apoptosis in fracture healing.	[117]
	*Bgn*	Depletion of miR-185 promotes osteogenic differentiation and suppresses bone loss in osteoporosis.	[118]
miR-128	*SIRT6*	Inhibition of miR-128 enhances osteogenic differentiation in osteoporosis.	[119]
miR-182	*FoxO1*	Antisense inhibition of miR-182 promotes osteoblast proliferation and differentiation, which consequently has positive effects on osteogenesis.	[120]
miR-877-5p	*EIF4G2*	Overexpression of miR-877-5p promotes osteogenic differentiation, which was characterized by increased cell mineralization, *ALP* activity and osteogenesis-related gene expression.	[121]
miR-224-5p	*Runx2* and *Sp7*	Inhibition of miR-224-5p promotes osteoblast differentiation and inhibits ovariectomy-induced bone loss and osteoporosis.	[122]
miR-433	*RUNX2*	Inhibits osteoblast differentiation by reducing *Runx2* transcript.	[123]
miR-132-3p	*Ep300*	Inhibits osteoblast differentiation; indirect effect on *Runx2* via *Ep300*, which regulates *Runx2* activity and stability.	[124]
miR-204	*RUNX2*	Impairs osteoblast differentiation & mineralization; promotes adipogenesis in MSCs; phytoestrogen Puerarin downregulates miR-204.	[125]
miR-383	*SATB2*	Negative regulator of osteogenesis.	[126]
	Directly targets *SATB2*; feedback with *Runx2*	Negative regulator; suppresses osteogenic differentiation.	[127]
miR-24-1-5p	*SMAD5*	Decreased osteoblast function; linked to reduced bone mass in maternal high-protein diet offspring.	[128]
miR-135-5p	*HIF1AN* (also predicted: *Smad5*)	Increased *ALP*, calcium deposition, *OPN*, *OCN*, and *Osx* promotes osteogenesis; but suppressed during *BMP2*-induced differentiation.	[129,130]
miR-129-5p	*TCF4* (*Wnt/β-catenin* transcription factor)	Overexpression suppresses osteoblast differentiation.	[131]
miR-409-5p	*LRP-8* (*Wnt* receptor)	Overexpression decreases osteoblast differentiation/mineralization.	[132]
miR-424-5p	*WIF1* (Wnt inhibitor)	Overexpression inhibits osteogenesis via *Wnt/β-catenin* suppression.	[133]
miR-378	*Sufu* (Hh pathway inhibitor)	Overexpression enhances osteogenesis & bone repair (rescues GC-induced ONFH).	[134]
miR-467g	*Runx2*, *Smo*, *Gli1* (*Hh* signaling)	Overexpression suppresses osteoblast proliferation and differentiation.	[135]
miR-151b	*Msx2* (transcription factor)	Suppresses osteogenesis and bone regeneration.	[136]
miR-487b-3p	*Nrarp*	Suppresses osteoblast differentiation; inhibition of miR-487b-3p improves bone formation and trabecular architecture in osteopenic mice.	[137]
miR-376c	*Wnt-3* and *ARF-GEF-1*	Suppresses osteoblast proliferation.	[138]
miR-593-5p	*LRP-6*	Impairs osteoblast differentiation, elevating bone resorption markers, and deteriorating trabecular bone architecture.	[139]
miR-139-3p	*ELK1*	Promotes osteoblast differentiation and suppresses apoptosis.	[140]
miR-3175	*DCAF1*	Suppresses apoptosis induced by dexamethasone (DEX).	[141]
miR-4523	*PGK1*	Suppresses apoptosis induced by DEX.	[142]
miR-7	*EGFR*	Promotes osteoblast apoptosis induced by DEX.	[143]
miR-107	*CAB39*	Promotes osteoblast apoptosis induced by DEX.	[144]
miR-455-3p	*HDAC2*	Promotes proliferation and inhibits apoptosis induced by ferric ammonium citrate (FAC).	[145]
miR-758-3p	*Caspase 3*	Inhibits apoptosis induced by FAC.	[146]
miR-3074-5p	*Smad4*	Promotes apoptosis induced by iron overload.	[147]
miR-1	*HSP-70*	Promotes apoptosis induced by nitric oxide.	[148]
miR-138	*TIMP1*	Promotes apoptosis (estrogen deficiency, H_2_O_2_).	[149]
miR-23a-3p	*PGC-1α*	Inhibits proliferation/differentiation and promotes apoptosis.	[150]
miR-342-5p	*BMP7*	Inhibits proliferation and promotes apoptosis.	[151]
miR-15b	*USP7*	Suppresses autophagy and differentiation.	[152]
miR-199a-3p	*IGF-1*, *mTOR*	Mediates osteoblast autophagy induced by estrogen.	[153]
miR-27a-3p	*CRY2*, *ERK1/2*	Promotes differentiation, suppresses apoptosis, and induces autophagy.	[154]

**Table 4 cells-14-01905-t004:** miRNAs governing both osteoclast and osteoblast activity.

miRNA	Cell Type	Role	Target Genes	Reference
miR-182 (Inhibits osteogenesis)	Osteoclast	Promotes differentiation and activity	*PKR*	[78]
	Osteoblast	Inhibits proliferation and differentiation	*FoxO1*	[120]
miR-214 (Inhibits osteogenesis)	Osteoclast	Promotes differentiation and activity	*PTEN*	[108]
	Osteoblast	Inhibits osteoblast activity and matrix mineralization	*ATF4*	[81]
miR-34a (Enhances osteogenesis)	Osteoclast	Inhibits differentiation	*RANKL and HIF1α*	[191,192]
	Osteoblast	Enhances differentiation	*NOTCH1 and DKK1*	[193,194]

**Table 5 cells-14-01905-t005:** Diagnostic potential of key circulating miRNAs in postmenopausal osteoporosis.

miRNA	Targets	Type of Biomarker	miRNA Source	Setting	Reference
miR-148a-3p	*MAFB*, *PPAR*, *WNT1*	Diagnostic biomarker distinguishing post-menopausal (PMO) from controls.	Serum	clinical	[199]
miR-375 and miR-532-3p	*ESR1*, *ADCY1*, *ATF2*, *CALM1*, *and PIK3R3*	Diagnostic biomarker distinguishing PMO from controls.	Serum	clinical	[201]
miR-21-5p, miR-23a-3p, and miR-125-5p	*PDCD4*, *ASL*, *EIF4E3* (miR-21), *RUNX2* (miR-23a-3p) *and PDGF* (miR-125-5p)	Diagnostic biomarker distinguishing PMO from controls.	Serum	clinical	[202]
miR-195 and miR-150	*GIT1*, *BMP* (miR-195), *and MMP14* (miR-150)	Biomarkers for diagnosing osteoporosis compared with controls.	Serum	clinical	[203]
Panel: miR-21, miR-23a, miR-24, miR-100, and miR-125b	*PDCD4* (miR-21) *RUNX2* (miR-23a, miR-24), *and BMPR2* (miR-100)	Biomarkers for diagnosing osteoporosis compared with controls.	Serum	clinical	[204]
miR-21 and miR-133a	*SPRY1* (miR-21)	Biomarkers for diagnosing osteopenia/osteoporosis compared with controls.	Plasma	clinical	[205]
Five miRNA panel comprising miR-30c-2-3p, miR-199a-5p, miR-424-5p, miR-497-5p, and miR-877-3p Four miRNA panel including miR-30c-2-3p, miR-199a-5p, miR-424-5p, and miR-877-3p	*HIF-1α pathway* (miR-199a-5p), *RUNX2* (miR-30c), *BMP signaling pathway* (miR-497), *Smad7 signaling* (miR-877-3p)	Diagnostic biomarker for osteoporosis vs. osteopenia and control.	Serum	clinical	[206]
miR-122-5p and miR-4516	*BMP2K*, *FSHB*, *IGF1R*, *RUNX2*, *SPARC*, *TSC22D3*, *TSC22D3*, *VDR (both) CNR2*, *ALPL*, *ANKH*, *ESR1*, *LRP6* (miR-122-5p), *CNR1*, *AR* (miR-4516)	Diagnostic biomarker for osteoporosis vs. osteopenia and control.	Serum and Plasma	clinical	[207]
miR-1246 and miR-1224-5p	*Tetraspanin 5* (miR-1224-5p)	Diagnostic biomarker for osteoporosis vs. osteopenia and control.	Plasma-exosomes	clinical	[208]
miR-140-3p and miR-23b-3p	*AKT1*, *AKT2*, *AKT3*, *BMP2 FOXO3*, *GSK3B*, *IL6R*, *PRKACB*, *RUNX2*, *WNT5*	Diagnostic biomarker for osteoporosis vs. osteopenia and control.	Serum	clinical	[209]
miR-23b-3p, miR-140-3p, miR-21-5p, miR-122-5p, and miR-125-5p	*SMAD7*, *FGF18*, *SKP2*, *SPRY1/2*, *PDCD4*, *PTEN*, *RECK*, *GDF-5*, *SOX2*, *PLAP1*, *ACVR2B* (miR-21- 5p) *RUNX2*, *MRC2*, *CCND1*, *PTEN* (miR23b-3p) *BMP2*, *IGF1R*, *RUNX2*, *SPARC*, *TSC22D3*, *VDR*, *PCP4* (miR-122-5p) *BMPR1B*, *TRAF6* (miR-125b-5p) *MCF2L*, *PTEN*, *BMP2* (miR-140-3p)	Biomarkers for differentiating osteoporotic and non-osteoporotic hip fractures.	Plasma	clinical	[210]
Panel for miR-203a, miR-31-5p, and miR-19b-1-5p	*FOS*, *RUNX2*, *SMAD1* (miR-203a) *PTEN*, *RUNX2* (miR19b-1-5p) *FZD3*, *RUNX2*, *SP7*, *SATB2* (miR-31-5p)	Biomarkers predicting fracture risk associated with diabetic osteoporosis.	Serum	clinical	[211]
miR-208a-3p, miR-155-5p, and miR-637	*ETS1*, *ACVR1* (miR208a-3p), *SOCS1* (miR-155-5p), *and SP7* (miR-637)	*miR-208a-3p* serves as a diagnostic biomarker for distinguishing PMO from both premenopausal osteoporosis and healthy controls, while *miR-155-5p* and miR-637 differentiate PMO from healthy controls.	Serum	clinical	[212]
miR-203a	*DLX5 and RUNX2*	Diagnostic marker of PMO vs. controls and monitoring marker of treatment response to zoledronate and teriparatide in PMO.		In vivo	[213]
OsteomiR™ panel	*RUNX2*, *LRP5*, *ß-catenin* (miR-375); *OCN*, *CTX* (miR-550a-3p); *WNT10B* (miR-152-3p); *BMP2*, *DLX5*, *OCN* (miR203a)	*miR-375* as a diagnostic biomarker for PMO vs. controls; *miR-203a* as a diagnostic biomarker for fragility fractures in postmenopausal osteoporosis vs. osteoporosis without fracture.	Serum	clinical	[200]
Panel for miR-152-3p, miR-30e-5p, miR-324-3p, miR-19b-3p, miR-335-5p, miR-19a-3p, miR550a-3p, miR-186-5p, miR-532-5p, miR-93-5p, miR-378-5p, miR-320a, miR-16-5p, miR-215-5p, let-7b-5p, miR-29b-3p, miR-7-5p, and miR365a-3p	*DKK1* (miR-152-3p, miR-335-3p); *LRP6* (miR-30e-5p); *BMP2* (miR-140-5p); *HDAC4*, *TGFß3; ACVR2A*, *CTNNBIP1*, *DUSP2* (miR-29b-2p)	Diagnostic biomarker of idiopathic and postmenopausal osteoporotic fractures vs. non-osteoporotic fractures.	Serum	clinical	[214]
miR-148a-3p	*WNT1*, *WNT10B*, *KDM6B*, *DNMT1*, *IGF1*, *MAFB*	Predictive biomarker of osteoporosis risk following acute spinal cord injury.	Plasma	clinical	[215]
Panel for miR-93-3p, miR-532-3p, miR-133a-3p, miR-301b-3p, miR-181c-5p, miR-203a-3p, miR-590-3p	*WNT1*, *LRP6*, *PTEN* (miR-301b); *DLX5*, *RUNX2* (miR203a-3p); *RANKL*, *MMP9*, *NF-ĸB*, *DKK1* (miR-218-5p); *DKK1* (miR-203)	Biomarker for X-linked primary osteoporosis.	serum	clinical	[216]
miR-124-3p, miR-2861, miR-21-5p, miR-23a-3p, miR-29a-3p	*SPRY*, *PDCD4* (miR-21), *RUNX2* (miR-23-3p)	Diagnostic biomarker of postmenopausal osteopenic/osteoporotic vertebral fractures vs. fracture-free postmenopausal controls.	Serum	In vivo	[217]

**Table 6 cells-14-01905-t006:** Delivery strategies to improve miRNA therapeutic stability, specificity, and efficacy.

Non-Virus-Based miRNA Delivery System			
Delivery System	Examples	Advantages	Disadvantages
Cationic lipoplexes	Cationic lipids with hydrophilic heads and hydrophobic tails forming complexes with nucleic acids [230].	Non-immunogenic, easy to manufacture, biocompatible.	Low efficiency and cytotoxicity.
Commercial lipoplexes	Lipofectamine^®^ RNAiMAX, SiPORT™ (Invitrogen, Waltham, MA, USA) [231,232] SilentFect™ (Bio-Rad, Hercules, CA, USA) [233].		
Polyethylene glycol (PEG)-modified liposomes	Cationic lipids conjugated with PEG [234]		
Polymeric delivery systems	Polyethyleneimines (PEIs) [235], low-molecular-weight PEIs [236], PEG/PEI conjugates [237,238], poly (lactic-co-glycolic acid) (PLGA) [239].	Non-immunogenic, transient expression, high packaging capacity.	Low gene-delivery efficiency in vivo, cytotoxicity.
Inorganic compound-based delivery systems	Gold nanoparticles (AuNPs) [240], PEG–gold nanoparticles [241], silica nanoparticles (SiNPs) [242].	High packaging capacity, non-immunogenic.	Low gene-delivery efficiency.
Extracellular vesicle-based delivery systems	miR-133b-enriched exosomes [243], miR-126-enriched exosomes [244], anti-miR-375–enriched BMSC exosomes [245].	High packaging capacity, non-immunogenic, tissue-specific delivery.	Large-scale EV production is challenging; there is a need for biogenesis control, immune profiling, and optimized administration routes.
**Virus-based miRNA delivery systems**			
Retroviral vectors (RVs)	Derived from lipid-enveloped RNA viruses [246] (e.g., Moloney murine leukemia virus, MoMLV) (miR-138 overexpression in murine embryonic fibroblasts) [247].	Stable transgene expression.	High carcinogenic potential due to insertional mutagenesis; cannot transduce non dividing cells.
Lentiviral vectors (LVs)	Lentivirus genus of *Retroviridae* family, including bovine (BIV), feline (FIV), equine (EIAV), simian (SIV), and human (HIV-2) immunodeficiency viruses [248,249,250,251].	Stable transgene expression, transduces dividing and non-dividing cells, broader tropism; long-term expression; lower insertional oncogenesis risk.	High chances of random genomic integrations which may cause insertional mutagenesis.
Adeno-Associated Virus (AAV) Vectors	AAV-based systems expressing artificial or therapeutic miRNAs, such as hemagglutinin-specific miRNAs [252] and miR-298 [253], have shown protective effects against influenza and neuromuscular diseases, respectively, in mice. AAV9 mediated suppression of *Shn3* to improve bone phenotype in osteoporotic mice [254].	Low immunogenicity, high transduction efficiency over a wide variety of cells, targeted tissue-specific gene expression based on serotype, infects dividing and non-dividing cells.	Low packaging capacity, production is tedious and expensive, immune activation is possible in large animals/humans; affected by promoter and serotype.

**Table 7 cells-14-01905-t007:** Role of miRNAs in Osteoporosis: Clinical Perspectives. A summary of clinical trials investigating the relationship between miRNAs and osteoporosis, with data retrieved from ClinicalTrials.gov.

Clinical Trial ID	NCT05912309	
	**Study title**	Effects of Time-restricted Eating and Exercise Training on Skeletal Muscle Mass Quantity, Quality and Function in Postmenopausal Women with Overweight and Obesity
	**Study type**	Interventional
	**Status**	Recruiting
	**Leading institution**	Public University of Navarra, Spain; Hospital of Navarra, Spain
	**Objective**	Assessing the effects of intermittent fasting and exercise on muscle mass, energy expenditure, cardiometabolic health, and miRNA biomarkers in postmenopausal women with obesity, menopause-associated conditions, sarcopenia, and osteoporosis.
	**Start and completion date**	Start date: 1 September 2023 Completion date (estimated): December 2025
	**Conditions**	Menopause-related conditions, Sarcopenia, Osteoporosis (Postmenopausal), Obesity
	**Link**	https://clinicaltrials.gov/study/NCT05912309?term=NCT05912309&rank=1 (8 February 2025)
**Clinical trial ID**	**NCT05556499**	
	**Study title**	The Bone-parathyroid Crosstalk in Primary Hyperparathyroidism (PARABONE)
	**Study type**	Observational
	**Status**	Not yet recruiting
	**Leading institution**	I.R.C.C.S. Ospedale Galeazzi-Sant’Ambrogio, Italy
	**Objective**	Examining the relationship between bone and parathyroid glands in primary hyperparathyroidism, with a focus on circulating lncRNAs and miRNAs in relation to bone metabolism.
	**Start and completion date**	Start date: October 2022 Completion date: 24 June 2026
	**Conditions**	Hyperparathyroidism (Primary), Osteoporosis, Parathyroid Neoplasms
	**Link**	https://clinicaltrials.gov/study/NCT05556499?term=NCT05556499&rank=1 (8 February 2025)
**Clinical trial ID**	**NCT05328154**	
	**Study title**	MAGnesium Effect with ANtiosteoporotic Drugs (MAGELLAN)
	**Study type**	Interventional
	**Status**	Completed
	**Leading institution**	University Hospital of Clermont-Ferrand, France
	**Objective**	Investigating whether a combination of bisphosphonates and magnesium is more effective than bisphosphonates alone in treating postmenopausal osteoporosis, with a focus on epigenetic biomarkers.
	**Start and completion date**	Start date: 28 June 2022 Completion date: 19 December 2024
	**Conditions**	Osteoporosis, Postmenopausal
	**Link**	https://clinicaltrials.gov/study/NCT05328154?term=NCT05328154&rank=1(8 February 2025)
**Clinical trial ID**	**NCT05421819**	
	**Study title**	Design and Development of a Novel Food Supplement for Osteoporosis Based on Gut Microbiome Mechanisms (OSTEOME)
	**Study type**	Interventional
	**Status**	Completed
	**Leading institution**	National and Kapodistrian University of Athens, Greece
	**Objective**	Examining the effects of a new food supplement based on gut microbiome mechanisms in osteoporosis patients, with an analysis of serum miRNA levels.
	**Start and completion date**	Start date: 15 June 2022 Completion date: 24 January 2024
	**Conditions**	Osteopenia, Postmenopausal osteopenia
	**Link**	https://clinicaltrials.gov/study/NCT05421819?term=NCT05421819&rank=1(8 February 2025)
**Clinical trial ID**	**NCT05228262**	
	**Study title**	Vascular Function, Sarcopenia and Pain in Postmenopausal Osteoporosis (VASCO)
	**Study type**	Interventional
	**Status**	Recruiting
	**Leading institution**	University Hospital, Clermont-Ferrand
	**Objective**	Evaluating the impact of osteoporosis treatments on cardiovascular health, sarcopenia, and pain, along with studying epigenetic biomarkers in osteoporotic women.
	**Start and completion date**	Start date: 15 February 2022 Completion date: 1 February 2027
	**Conditions**	Osteoporosis, Postmenopausal
	**Link**	https://clinicaltrials.gov/study/NCT05228262?term=NCT05228262&rank=1#study-overview (8 February 2025)
**Clinical trial ID**	**NCT05673837**	
	**Study title**	The Type ONe dIabetic Bone Collaboration Study (TONICS)
	**Study type**	Observational
	**Status**	Active, not recruiting
	**Leading institution**	Odense University Hospital, Denmark
	**Objective**	Analyzing bone health in individuals with Type 1 diabetes, focusing on miRNAs related to bone metabolism and osteoporosis epidemiology.
	**Start and completion date**	Start date: 10 December 2021 Completion date: 1 April 2024
	**Conditions**	Osteoporosis (Secondary), Diabetes Mellitus (Type 1)
	**Link**	https://clinicaltrials.gov/study/NCT05673837?term=NCT05673837&rank=1 (8 February 2025)
**Clinical trial ID**	**NCT03931109**	
	**Study title**	Circulating miRNA in Primary Hyperparathyroidism
	**Study type**	Observational [Patient Registry]
	**Status**	Active, not recruiting
	**Leading institution**	University of Pennsylvania, Pennsylvania
	**Objective**	Investigating serum miRNA expression levels in patients with primary hyperparathyroidism, both with and without osteoporosis.
	**Start and completion date**	Start date: 7 September 2018 Completion date: 1 July 2024
	**Conditions**	Primary Hyperparathyroidism, Osteoporosis (Postmenopausal)
	**Link**	https://clinicaltrials.gov/study/NCT03931109?term=NCT03931109&rank=1(8 February 2025)
**Clinical trial ID**	**NCT02128009**	
	**Study title**	Study on the microRNA Expression Level in Postmenopausal Osteoporosis (microRNA)
	**Study type**	Observational [Patient Registry]
	**Status**	Completed
	**Leading institution**	Fujian Academy of Traditional Chinese Medicine, China
	**Objective**	Studying molecular mechanisms by analyzing miRNA levels in postmenopausal osteoporosis patients with kidney yin deficiency syndrome
	**Start and completion date**	Start date: February 2014 Completion date: January 2017
	**Conditions**	Postmenopausal Osteoporosis
	**Link**	https://clinicaltrials.gov/study/NCT02128009?term=NCT02128009&rank=1(8 February 2025)
**Clinical trial ID**	**NCT03472846**	
	**Study title**	MiDeTe-microRNA Levels Under Denosumab and Teriparatide Therapy in Postmenopausal Osteoporosis
	**Study type**	Interventional
	**Status**	Completed
	**Leading institution**	Medical University of Vienna, Austria
	**Objective**	Measuring bone-specific miRNA levels in the serum of 26 postmenopausal women undergoing antiresorptive or osteoanabolic osteoporosis treatments
	**Start and completion date**	Start date: March 2017 Completion date: 30 September 2022
	**Conditions**	Postmenopausal Osteoporosis, Diabetes Type 2
	**Link**	https://clinicaltrials.gov/study/NCT03472846?term=NCT03472846&rank=1(8 February 2025)
**Clinical trial ID**	**NCT02705040**	
	**Study title**	Roles of microRNAs in the Development of Osteoporosis in Men-Preliminary Study
	**Study type**	Observational
	**Status**	Unknown status
	**Leading institution**	Taipei Medical University WanFang Hospital, Taiwan
	**Objective**	Observational study exploring the role of specific miRNAs in men with osteoporosis.
	**Start and completion date**	Start date: December 2013 Completion date: December 2016
	**Conditions**	Osteoporosis
	**Link**	https://clinicaltrials.gov/study/NCT02705040?term=NCT02705040&rank=1(8 February 2025)
**Clinical trial ID**	**NCT01875458**	
	**Study title**	Biomarker Identification in Orthopedic & Oral Maxillofacial Surgery Subjects to Identify Risks of Bisphosphonate Use
	**Study type**	Observational
	**Status**	Completed
	**Leading institution**	University of Pennsylvania, Austria
	**Objective**	Identifying DNA and miRNA biomarkers that influence metabolic response to bisphosphonate treatment
	**Start and completion date**	Start date: 13 March 2012 Completion date: 30 October 2022
	**Conditions**	Osteoporosis, With or Without Treatment Bisphosphonate Treatment Atypical Femur Fracture, Bisphosphonate Related Osteonecrosis of the Jaws (BRONJ), Healthy Volunteers
	**Link**	https://clinicaltrials.gov/study/NCT01875458?term=NCT01875458&rank=1(8 February 2025)

## Data Availability

During the preparation of this work the authors used ChatGPT (GPT-5.1) in order to improve the readability and language of the manuscript. After using this tool, the authors reviewed and edited the content as needed and take full responsibility for the content of the published article.

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
