# Peer review of "The Growing Significance of microRNAs in Osteoporosis"

_cells, 2025, doi:10.3390/cells14231905_

Round 1
Reviewer 1 Report
Comments and Suggestions for Authors
The manuscript entitled “The Growing Significance of microRNAs in Osteoporosis” is a narrative review that gathers data from numerous studies about the role of miRNAs in osteoblasts and osteoclasts, as well as their potential in diagnosis and therapeutics. The topic is relevant, and the manuscript presents an adequate structure. However, in the reviewer’s opinion, it is also unnecessarily lengthy. For instance, Osteoporosis is defined at least twice, as well as the osteoblast and osteoclast origins and functions. Moreover, Table 2 is described twice, before and after its appearance. These repetitions are unnecessary and make the reading fatiguing. Below, the authors can find some suggestions and observations.
Abstract:
Adequate structure is present, with a balanced distribution of information between general and bone-focused sections. As a minor suggestion, the definition of miRNAs could be summarized or removed.
Introduction
Overall, this section is well-written and follows an adequate line of thinking. However, it is lengthy. Some sentences are repeated (i.e., osteoporosis definition and characteristics), and some parts are overwhelmingly detailed for an introduction. For instance, lines 119 – 125 and 136 – 143 could be significantly summarized. Moreover, the examples and signaling pathways related to miRNAs in bone can be detailed in subsequent sections. Minor observations can be found below.
Lines 43 and 44: Revise the sentence and punctuation to enhance clarity.
Line 52: Consider replacing “regeneration” with “turnover.”
Lines 57 to 64: Osteocytes were not mentioned. Since their role in orchestrating bone remodeling is crucial, the reviewer suggests including their relevance in this section.
Lines 73 – 75: Consider adding the term “density” to osteoporosis’s description to align with the medication’s role.
Line 155: consider adding “also” after but.
Bone physiology:
Gene names should be written in italic.
Include the meaning of NFATc1 when using the abbreviation for the first time.
Add the abbreviation TRAP in line 216.
Lines 218 – 224: These sentences are confusing, mixing inhibition and promotion of osteoclastogenesis. Moreover, since WNT, FGF, and TGF-β signaling are mentioned as osteoblast-differentiation pathways, their dual role should be further developed or removed from this section.
Lines 226 – 234: Add more references or distribute them.
Figure 1, minor suggestions to improve clarity:
- Add some osteocytes to the first resting bone surface
- Change the term “mononuclear cells” to a more specific one (e.g., BMMSCs) to avoid confusion with blood-derived monocytes.
- Add (can be in the figure’s legend) the function of the “mononuclear cells”
- Add some osteoblasts inside the osteoid
Lines 282 – 292: The definition and characterization of osteoporosis are already described in the introduction. Therefore, this information should be grouped in just one of these sections 2.2 or introduction) and greatly summarized in the other.
Section 3:
Add the meaning of * in line 355
Figure 3’s legend: The information is already in the text; therefore, it can be removed or summarized in the legend.
Lines 391 – 394: Revise the sentence; it seems unfinished. Is the inhibition of miRNAs that causes the phenotypes or any type of alteration?
In the reviewer’s opinion, the effects observed in humans are not “in vivo effects” but clinical ones. Thus, consider revising the table 1 legend or column 3 title.
Tables 2, 3, and 4 are the most important syntheses of the present review and were properly devised. As the only suggestion, add the meaning of the abbreviations that cannot be found in the text as a footnote immediately below each table.
As the reviewer sees, the authors should discuss whether the osteoclast-targeted inhibition by miRNAs would produce the same adverse effects found in the use of bisphosphonates and denosumab (i.e., bone necrosis and bone matrix aging), as well as their potential to be employed for inhibiting bone metastases or osteosarcomas.
Subsection 4.3: The clarity of the text can be improved. In some examples of miRNAs, only one of the dual functions appeared to be described. Additionally, in other examples, the reviewer believes that the authors described the inhibition of certain miRNAs (i.e., MiR-324) to achieve an inhibitory or promotional effect, which is the opposite of the miRNA function. One suggestion to improve it is to begin and conclude the description of miRNAs that inhibit osteogenesis, followed by descriptions of miRNAs that promote osteogenesis. Same as for Table 4.
Section 5:
Consider splitting this section into “5.1. diagnosis” and “5.2 therapeutics” and creating a table or a scheme with the main techniques employed to improve their delivery, specificity to target tissues, and stability.
In Table 5’s legend, specify where these miRNAs were observed (in vitro, in vivo, or clinical settings). If the studies are heterogeneous in this matter, add a column to specify for each one.
Consider adding some discussion of local delivery of miRNAs vs. systemic administrations to overcome the off-target effects observed.
Comments on the Quality of English LanguageMinor revisions necessary.
Author Response
Reviewer 2’s comments:
The manuscript entitled “The Growing Significance of microRNAs in Osteoporosis” is a narrative review that gathers data from numerous studies about the role of miRNAs in osteoblasts and osteoclasts, as well as their potential in diagnosis and therapeutics. The topic is relevant, and the manuscript presents an adequate structure. However, in the reviewer’s opinion, it is also unnecessarily lengthy. For instance, Osteoporosis is defined at least twice, as well as the osteoblast and osteoclast origins and functions. Moreover, Table 2 is described twice, before and after its appearance. These repetitions are unnecessary and make the reading fatiguing. Below, the authors can find some suggestions and observations.
- We thank the reviewer for constructive comments. As suggested, we removed the repetitive and unnecessary sentences and modified Table 2 in the revised manuscript.
Abstract: Adequate structure is present, with a balanced distribution of information between general and bone-focused sections. As a minor suggestion, the definition of miRNAs could be summarized or removed.
- As suggested by the reviewer, we removed the definition of microRNAs and revised the abstract to improve clarity, focus, and presentation of the key findings (line17-33).
Introduction: Overall, this section is well-written and follows an adequate line of thinking. However, it is lengthy. Some sentences are repeated (i.e., osteoporosis definition and characteristics), and some parts are overwhelmingly detailed for an introduction. For instance, lines 119 – 125 and 136 – 143 could be significantly summarized. Moreover, the examples and signaling pathways related to miRNAs in bone can be detailed in subsequent sections. Minor observations can be found below.
- As suggested by the reviewer, we have shortened the introduction section by removing repetitive sentences, summarizing overly detailed parts, and relocating examples and signaling pathways related to miRNAs in bone to the relevant subsequent sections.
Lines 43 and 44: Revise the sentence and punctuation to enhance clarity.
- As suggested by the reviewer, this sentence was revised (lines 40-43).
Line 52: Consider replacing “regeneration” with “turnover.”
- Thank the reviewer for the suggestion. We replaced the “regeneration” with “turnover” throughout the revised manuscript.
Lines 57 to 64: Osteocytes were not mentioned. Since their role in orchestrating bone remodeling is crucial, the reviewer suggests including their relevance in this section.
- As suggested by the reviewer, we included the role of osteocytes in the revised manuscript (lines 59-64).
Lines 73 – 75: Consider adding the term “density” to osteoporosis’s description to align with the medication’s role.
- As suggested by the reviewer, we added “density” to osteoporosis’s description in the revised manuscript (line 65).
Line 155: consider adding “also” after but.
- Thank the reviewer for the correction.
Bone physiology:
Gene names should be written in italic.
- All gene names were changed to italic throughout the revised manuscript.
Include the meaning of NFATc1 when using the abbreviation for the first time.
- Thank the reviewer for the correction (line 354).
Add the abbreviation TRAP in line 216.
- Thank the reviewer for the correction (line 421)
Lines 218 – 224: These sentences are confusing, mixing inhibition and promotion of osteoclastogenesis. Moreover, since WNT, FGF, and TGF-β signaling are mentioned as osteoblast-differentiation pathways, their dual role should be further developed or removed from this section.
- We agree with the reviewer’s concern that the current sentences are confusing due to the simultaneous mention of both inhibitory and promotive effects on osteoclastogenesis. We revised the text to clearly distinguish these opposing roles and ensure the narrative is consistent (line 133-144).
Lines 226 – 234: Add more references or distribute them.
- As suggested by the reviewer, we distributed references.
Figure 1: minor suggestions to improve clarity.
- Add some osteocytes to the first resting bone surface
- Osteocytes were added to the revised Figure 1.
- Change the term “mononuclear cells” to a more specific one (e.g., BMMSCs) to avoid confusion with blood-derived monocytes.
- We changed “mononuclear cells” to “BMMSCs”.
- Add (can be in the figure’s legend) the function of the “mononuclear cells”
- The function of the “mononuclear cells” was added to the revised Figure 1.
- Add some osteoblasts inside the osteoid
- Osteoblasts were added to inside of the osteoid.
Lines 282 – 292: The definition and characterization of osteoporosis are already described in the introduction. Therefore, this information should be grouped in just one of these sections 2.2 or introduction) and greatly summarized in the other.
- As suggested by the reviewer, we removed the repetitive sentences describing the definition and characterization of osteoporosis from the Section 2.2. and presented only once in introduction section (lines 65-105).
Section 3:
Add the meaning of * in line 355
- Thank the reviewer for the correction (line 290).
Figure 3’s legend: The information is already in the text; therefore, it can be removed or summarized in the legend.
- As suggested by the reviewer, we summarized the information in the revised Figure 3 legend to avoid redundancy (lines 298-303).
Lines 391 – 394: Revise the sentence; it seems unfinished. Is the inhibition of miRNAs that causes phenotypes or any type of alteration?
- We thank the reviewer for bringing this point to our attention. It refers to any alteration that results in the phenotypes described in Table 1. Further, the sentences were revised as suggested by the reviewer (line 324-330).
In the reviewer’s opinion, the effects observed in humans are not “in vivo effects” but clinical ones. Thus, consider revising table 1 legend or column 3 title.
- We changed the legend of Table 3 to “Effects of miRNA deletions or mutations on skeletal development” and updated the column title to “Effect on skeletal development”.
As the reviewer sees, the authors should discuss whether the osteoclast-targeted inhibition by miRNAs would produce the same adverse effects found in the use of bisphosphonates and denosumab (i.e., bone necrosis and bone matrix aging), as well as their potential to be employed for inhibiting bone metastases or osteosarcomas.
- Thank the reviewer for pointing this out. Unfortunately, we were unable to identify any comparative studies demonstrating that osteoclast-targeted miRNA inhibition has adverse effects analogous to bisphosphonates or denosumab—such as bone necrosis or bone matrix aging. Moreover, there is currently insufficient evidence regarding their potential use in preventing bone metastases or osteosarcomas. In the absence of such data, it is difficult to incorporate a meaningful discussion of these aspects into the revised manuscript.
Subsection 4.3: The clarity of the text can be improved. In some examples of miRNAs, only one of the dual functions appeared to be described. Additionally, in other examples, the reviewer believes that the authors described the inhibition of certain miRNAs (i.e., MiR-324) to achieve an inhibitory or promotional effect, which is the opposite of the miRNA function. One suggestion to improve it is to begin and conclude the description of miRNAs that inhibit osteogenesis, followed by descriptions of miRNAs that promote osteogenesis. Same as for Table 4.
- Thank the reviewer for intuitive suggestion. We separated miRNAs inhibiting osteogenesis from miRNAs promoting osteogenesis in the revised manuscript. Additionally, we revised the sentences describing inhibition of a miRNA as its direct function (line 456-473). Finally, we revised Table 4 as suggested by the reviewer.
Section 5:
Consider splitting this section into “5.1. diagnosis” and “5.2 therapeutics” and creating a table or a scheme with the main techniques employed to improve their delivery, specificity to target tissues, and stability.
- As suggested by the reviewer, we divided into two subsections: “5.1-Diagnostic applications of miRNAs in osteoporosis” and “5.2- Therapeutic applications of miRNAs in osteoporosis”. Additionally, we added a new table 6 “Delivery strategies to improve miRNA therapeutic stability, specificity, and efficacy” to the revised manuscript.
In Table 5’s legend, specify where these miRNAs were observed (in vitro, in vivo, or clinical settings). If the studies are heterogeneous in this matter, add a column to specify for each one.
- Thank the reviewer for pointing this out. We added a new column “setting,” that indicates whether each reported effect was observed in a clinical setting or an in vivo setting.
Consider adding some discussion of local delivery of miRNAs vs. systemic administrations to overcome the off-target effects observed.
- As suggested by the reviewer, we added the sentences comparing local versus systemic delivery of miRNAs, highlighting how local administration can help minimize off-target effects while improving therapeutic specificity (lines 576-581).

Reviewer 2 Report
Comments and Suggestions for Authors
This manuscript provides a comprehensive and up-to-date review of the emerging roles of microRNAs in osteoporosis, focusing on their regulation of osteoblast and osteoclast differentiation and potential as therapeutic targets. The topic is timely and relevant to the readership of Cells, and the review is supported by an extensive list of references.
However, the manuscript is quite long and contains sections that read more like a textbook than a focused scientific review. I recommend the following revisions to enhance clarity and impact:
-
Condense Section 2 (Bone Physiology) to briefly summarize essential background and shift the main emphasis toward the miRNA sections (Sections 4–5).
-
Clarify the novelty of this review in the Introduction and Conclusion—e.g., highlight recent findings on miRNA-mediated osteoblast–osteoclast communication (such as exosomal miR-214 signaling).
-
Improve figure and table consistency, particularly Figure 4 (interaction diagram) and Tables 1–3 (formatting of gene/miRNA names and alignment).
-
Add brief discussion of translational aspects, including clinical or preclinical miRNA-based therapies currently in development.
-
Correct typographical and formatting errors (e.g., “Ra us ra us” → Rattus rattus; “be er” → better), and standardize terminology for genes (e.g., RUNX2, DKK1) and miRNAs (e.g., miR-214-3p).
Overall, this is a scientifically sound and valuable review that will be suitable for publication after moderate revision and editorial polishing.
Comments on the Quality of English LanguageThe English is generally clear and professional, but the manuscript would benefit from minor editing by a native or professional scientific editor to enhance fluency and conciseness.
Several sentences are overly long or repetitive, and simplifying these would improve readability. Consistent use of tense, technical terminology, and formatting (especially for gene and miRNA names) is also recommended.
Author Response
Reviewer 3’s comments:
This manuscript provides a comprehensive and up-to-date review of the emerging roles of microRNAs in osteoporosis, focusing on their regulation of osteoblast and osteoclast differentiation and potential as therapeutic targets. The topic is timely and relevant to the readership of Cells, and the review is supported by an extensive list of references.
However, the manuscript is quite long and contains sections that read more like a textbook than a focused scientific review. I recommend the following revisions to enhance clarity and impact:
- We thank the reviewer for the positive feedback and for recognizing the significance of our manuscript. We appreciate the constructive suggestions, which we believe have helped improve and strengthen the revised version.
- Condense Section 2 (Bone Physiology) to briefly summarize essential background and shift the main emphasis toward the miRNA sections (Sections 4–5).
- We revised section 2 as suggested by the reviewer (line 163-246).
- Clarify the novelty of this review in the Introduction and Conclusion—e.g., highlight recent findings on miRNA-mediated osteoblast–osteoclast communication (such as exosomal miR-214 signaling).
- As suggested by the reviewer, we highlighted recent findings on miRNA-mediated osteoblast–osteoclast communication in the revised introduction section (lines 145-155).
- Improve figure and table consistency, particularly Figure 4 (interaction diagram) and Tables 1–3 (formatting of gene/miRNA names and alignment).
- Thank the reviewer for pointing this out. We improved consistency of Figures and Tables in the revised manuscript.
- Add brief discussion of translational aspects, including clinical or preclinical miRNA-based therapies currently in development.
- As suggested by the reviewer, we added brief discussion of translational aspects of miRNAs to the revised Tables 5 and 7.
- Correct typographical and formatting errors (e.g., “Ra us ra us” → Rattus rattus; “be er” → better), and standardize terminology for genes (e.g., RUNX2, DKK1) and miRNAs (e.g., miR-214-3p).
- Thank the reviewer for the correction.
Comments on the Quality of English Language: The English is generally clear and professional, but the manuscript would benefit from minor editing by a native or professional scientific editor to enhance fluency and conciseness. Several sentences are overly long or repetitive and simplifying these would improve readability. Consistent use of tense, technical terminology, and formatting (especially for gene and miRNA names) is also recommended.
- Our revised manuscript has been edited by a professional scientific editor of our department for grammar and overall language fluency.

Round 2
Reviewer 1 Report
Comments and Suggestions for Authors
The revised manuscript, entitled “The Growing Significance of microRNAs in Osteoporosis,” has undergone several improvements and additions that enhance the review. The authors have addressed all comments and concerns raised by the reviewer and exceeded expectations, including the addition of better schemes and more tables. The reviewer congratulates the authors and has no further suggestions.
Comments on the Quality of English LanguageThe language has been improved. Minor revisions can be performed during the proofreading process.
Author Response
The revised manuscript, entitled “The Growing Significance of microRNAs in Osteoporosis,” has undergone several improvements and additions that enhance the review. The authors have addressed all comments and concerns raised by the reviewer and exceeded expectations, including the addition of better schemes and more tables. The reviewer congratulates the authors and has no further suggestions.
Author response: We thank the reviewer for constructive comments